# Alginate-Based Bio-Composites and Their Potential Applications

**DOI:** 10.3390/jfb13030117

**Published:** 2022-08-10

**Authors:** Khmais Zdiri, Aurélie Cayla, Adel Elamri, Annaëlle Erard, Fabien Salaun

**Affiliations:** 1Laboratoire de Génie et Matériaux Textiles, École Nationale Supérieure des Arts et Industries Textiles, Université de Lille, 59000 Lille, France; 2Laboratoire de Physique et Mécanique Textiles, École Nationale Supérieure d’Ingénieurs Sud-Alsace, Université de Haute Alsace, EA 4365, 68100 Mulhouse, France; 3Unité de Recherche Matériaux et Procédés Textiles, École Nationale d’Ingénieurs de Monastir, Université de Monastir, UR17ES33, Monastir 5019, Tunisia

**Keywords:** alginate, fiber, thermo-mechanical, physico-chemical, wound-healing

## Abstract

Over the last two decades, bio-polymer fibers have attracted attention for their uses in gene therapy, tissue engineering, wound-healing, and controlled drug delivery. The most commonly used bio-polymers are bio-sourced synthetic polymers such as poly (glycolic acid), poly (lactic acid), poly (e-caprolactone), copolymers of polyglycolide and poly (3-hydroxybutyrate), and natural polymers such as chitosan, soy protein, and alginate. Among all of the bio-polymer fibers, alginate is endowed with its ease of sol–gel transformation, remarkable ion exchange properties, and acid stability. Blending alginate fibers with a wide range of other materials has certainly opened many new opportunities for applications. This paper presents an overview on the modification of alginate fibers with nano-particles, adhesive peptides, and natural or synthetic polymers, in order to enhance their properties. The application of alginate fibers in several areas such as cosmetics, sensors, drug delivery, tissue engineering, and water treatment are investigated. The first section is a brief theoretical background regarding the definition, the source, and the structure of alginate. The second part deals with the physico-chemical, structural, and biological properties of alginate bio-polymers. The third part presents the spinning techniques and the effects of the process and solution parameters on the thermo-mechanical and physico-chemical properties of alginate fibers. Then, the fourth part presents the additives used as fillers in order to improve the properties of alginate fibers. Finally, the last section covers the practical applications of alginate composite fibers.

## 1. Introduction

There is a great variety of bio-polymers, including the family of polysaccharides such as sodium alginate, which are produced from marine products. These bio-polymers constitute an interesting alternative for replacing polymers derived from petrochemicals, as they have important physico-chemical and biological properties [1,2]. Alginate bio-polymers can form a reticulated structure when they link with chloride ions, Na^+^, or calcium ions, Ca^2+^ [3]. These properties make these bio-polymers very useful materials in various fields, including water treatment, packaging, textiles, agriculture, pharmaceuticals, electronics, and the biomedical field [4,5].

This bio-polymer has been exploited in several forms. Among these forms, fibers, nano-fibers, micro-spheres, micro-particles, and nano-particles have been studied. Many authors have spun alginate fibers using a variety of methods: electro-spinning, the microfluidic system, and the traditional wet-spinning technique [6]. Wet-spinning is a technique that can be easily scaled up and is commonly used in high-performance fiber industries to create materials such as aramid fibers. This spinning technique is a higher throughput, more viable and cost-effective technique than the electro-spinning method [7].

However, there are limitations for the use of calcium alginate fibers. Indeed, alginate fibers have a low mechanical strength and poor thermal stability [8]. Additionally, the contact of alginate with the physiological environment makes this bio-polymer inappropriate for uses related to load-bearing body parts [9]. The incorporation of adhesive peptide and natural or synthetic polymers into alginate fibers gives them their desired properties and allows them to be used in innovative processes. Indeed, alginate fiber composites have shown a strong development potential, particularly in biological, medical, electronics, and food packaging applications [10].

In the present review, the structure and specific properties of alginate are described, followed by the discussion of the different strategies of alginate spinning and the impact of the process and solution parameters on the properties of these fibers. A comprehensive study of literature works on the incorporation of a wide range of materials into alginate fibers is carried out. Additionally, the exploitation and exploration of bio-composite fibers of alginate in several applications are discussed. This review can be used as a reference for researchers and industrials in their potential works.

## 2. Definition, Source, and Structure of Alginate

Alginate is a relatively abundant natural polysaccharide, since it is the structural component of brown algae. It is mainly generated from *Laminaria hyperborea*, *Laminaria digitata*, *Laminaria japonica*, *Ascophyllum*, and *Macrocystis pyrifera* [11].

Nowadays, this polysaccharide is used in many sectors, thinks to its unique colloidal properties, which means that it can be used as a thickener, stabilizer, film-forming agent, gelling agent, etc. [12].

The extraction of alginate from algae is based on the water solubility of this polymer. Alginic acid is insoluble in water, whereas sodium Na^+^ or potassium K^+^ salts are soluble. It is therefore necessary to convert all alginate salts (calcium and potassium) into sodium alginate salts using sodium carbonate. The main steps of the alginate extraction process are [13,14]:✓Pre-treatment: the algae are washed several times with water and then rinsed with distilled water in order to remove any impurities. The algae are then dried and finely crushed.✓Purification: the seaweed powder is treated with a dilute solution of acid, capable of dissolving sugars. This causes the formation of alginic acid.✓Extraction: the alginic acid is redissolved in a slightly basic solution of sodium carbonate NaHCO_3_ (concentration 1.5%) at temperatures over a range of 50–90 °C for 1–2 h. This converts the alginic acid into sodium alginate.✓Recovery: The sodium alginate, which is not soluble in a mixture of alcohol and water, can be separated from the system. Indeed, the sodium alginate solution is then filtered and an addition of ethanol allows for the precipitation of the alginate. The process ends with a drying and grinding step to obtain a powder with the appropriate particle size.

Alginates have specific characteristics that differ from other natural polymers [15]. They are mainly composed of two uronic acid residues: β-D-mannuronic acid (M block) and α-L-guluronic acid (G block). These two acid units are linked by β-1,4 glycosidic bonds [16] according to a sequence of three types of blocks. The latter can be homo-polymer blocks (-M-M-M- or -G-G-G-) or co-polymer blocks (-M-G-M-) (Figure 1).

It is important to note that the distribution (poly-dispersity) and the length of the blocks depend on the species and the part of the algae from which the alginic acid is extracted.

Alginate is characterized by several properties such as low cost, non-toxicity, bio-compatibility, bio-degradability, liquid absorption capacity, non-immunogenicity, abundance of availability in nature, and ease of chemical derivatization. These properties make alginate one of the most commonly used bio-polymers over a wide application area, such as a thickener in many food products, pharmaceutical formulations, tissue engineering, cell culture, and textile printing agents [17].

## 3. Properties of Alginate

### 3.1. Physico-Chemical Behaviors of Alginates

Pure alginic acid is insoluble in water. Its solubility or not in water depends on the type of salts associated with it. The sodium, ammonium, and potassium salts dissolve perfectly in aqueous solution, giving highly viscous solutions. In contrast, the magnesium and calcium salts are insoluble.

Sodium alginate is soluble in water at low ionic strength values. The solubilization of the polyanion becomes slower with an increase in the salinity of the medium.

The pH of the solution and the ionic strength of the solvent play an important role in the solubility of the alginate. Indeed, phase separation or even the formation of a hydrogel may occur when the pH of the solution is below the pKa of guluronic acid (pKa = 3.65) and mannuronic acid (pKa = 3.38) [18]. The rate of dissolution of the alginate decreases with an increase in the ionic strength [19].

Sodium alginate can have a stability of several months, and can be stored in a dry and cool place away from sunlight. At low temperatures, sodium alginate can be stored for several years without a significant reduction in its molecular weight. On the other hand, dry alginic acid has a very limited stability at ordinary temperatures due to its intramolecular degradation [20].

In the face of their multiple uses, it is important to be aware of the factors that determine and limit the stability of aqueous alginate solutions, and the chemical reactions responsible for their degradation.

The relative viscosity of an alginate solution can be severely reduced over a short period of time under conditions favoring degradation.

Alginates are increasingly being used because of their ability to form gels, and more precisely, hydrogels. There are two types of hydrogels: chemical hydrogels and physical hydrogels.

The nodes of the network are covalent bonds, in the case of chemical hydrogels. This type of hydrogel is irreversible, due to the irreversible nature of the covalent bond [21]. Physical hydrogels are formed by chains entanglements, hydrogen bonds, ionic bonds, and van der Waals interactions. These physical interactions have low energy, allowing the reversibility of the hydrogels [22].

Alginates have the ability to form a physical hydrogel through inter-chain interactions [23]. This interaction is described by the egg-box model, in which each divalent ion can interact with two adjacent units, G, or those belonging to two opposite chains [24].

The salt of the divalent cation that is generally used for alginate gelling is calcium chloride, because of its good solubility in aqueous medium, and the high availability of the calcium ions. The latter are retained in a kind of cage cavity, and interact with the carboxylate functions and oxygen atoms of the hydroxyl functions (Figure 2) [25,26].

The properties of alginate gel formation are influenced by the molecular weight, divalent ions, and chemical structure.
Me^2+^ = Ca^2+^, Zn^2+^, Cu^2+^

### 3.2. Biological Properties of Alginate

A biocompatible material is one that is capable of not degrading the biological environment in which it is used [27]. The alginate composition has a very important role on its biocompatibility properties. In fact, alginates with a high M content have been reported to be immunogenic and more potent in inducing cytokine production compared to alginates with a high G content.

The immunogenic response during an alginate injection or at the implantation site could be attributed to the various impurities remaining in the alginate during its extraction. Nevertheless, the studies of Orive et al. [28], and Jangwook and Kuen Yong Lee [29] have shown that the purification of alginate did not induce an immunogenic response in animals.

Moreover, this natural substance has the advantage of having a hemostasis property. The exchange between sodium Na^+^ ions present in the wound exudate and calcium Ca^2+^ ions will form a non-adherent hydrophilic gel that fills the wound and creates a microclimate favorable for healing [30].

Additionally, calcium alginate has healing properties. It has been described that alginate fibers are able to form a gel, which allows for the creation of a moist microenvironment favorable to the scarring process. Therefore, it has a high absorption capacity, superior to those of hydro-colloids and hydro-cellulars [31,32]. It has been shown that alginates rich in mannuronic units have positive impacts on cicatrization, while other authors believe that this activity also depends on the level of purification of the alginate [33,34].

The high water absorption capacity of alginate confirms its use at all wound-healing stages for highly exudative wounds [35].

The intra-fiber absorption of alginate allows for the textile support used for wound-healing, to limit the risk of wound infection. Indeed, the swelling of the fibers via the absorption of fluids facilitates the solubilization of the alginate, which makes it possible to immobilize the bacteria present on the textile.

The absorption of wound fluids by the calcium alginate compress allows for the formation of a highly viscous solution via ion exchange between the calcium of the alginate and the sodium present in the blood [36]. This moist microenvironment promotes healing.

## 4. Production of Alginate Fibers

### 4.1. Process

Fibers of alginate have attracted extensive interest because of their ease of handling, high surface area, and its ability to retain mechanical integrity [37].

The melt spinning process is not adopted for alginates. In fact, the latter are characterized by several hydrogen bonds, responsible for a glass transition and a melting temperature above their thermal decomposition temperature. Alginate fibers and alginate composite fibers are elaborated using several techniques: the wet-spinning method, electro-spinning, and the micro-fluidic system.

#### 4.1.1. Fibers Formation via Wet-Spinning

The most commonly used spinning technique for alginates is wet-spinning (Figure 3) [38].

The alginate powder can be dissolved in distilled water. After degassing, the sodium alginate solution can be extruded through a die immersed into a coagulation bath containing calcium chloride (CaCl_2_) solution, to cross-link the alginate.

An interaction is performed between the Ca^2+^ cations and the negatively charged molecule of alginate, promoting the solidification of alginate networks (Figure 4) [39].

After an appropriate immersion time, the calcium alginate filaments can be washed in a bath of demineralized water. These obtained filaments can then be stretched and dried at a temperature of 80 °C [40].

#### 4.1.2. Fibers Formation via Electro-Spinning

Electro-spinning is a highly versatile spinning process from a technical point of view. Figure 5 represents the principle diagram of the electrostatic alginate spinning device. This device usually consists of a high-voltage supply system, a syringe with a metal needle, a syringe pump, and a target or collector system [41].

Compared to the wet-spinning method, electro-spinning allows for the elaboration of ultrathin alginate fibers by placing a bio-polymer in between the charged electrode. When a strong electric field is applied, generally in the order of kilovolts (kV), the molten polymer is ejected towards the collector electrode. This allows for the formation of a fibrous mat consisting of individual fibers with nanometer- to micrometer-size diameters [42].

The electro-spinning technique of alginate has a large application in the biomedical domain because it can provide a larger specific surface area than the traditional wet-spinning technique [43].

However, it is difficult to manufacture continuous and uniform electro-spun fibers from neat alginate solutions using the method of electro-spinning, due to the rigid structure of the alginate and the lack of molecular entanglement.

#### 4.1.3. Fibers Formation via the Microfluidic System

The microfluidic system gives precise control of the morphology and the dimensional characteristics of the elaborated fiber. This is attributed to the formation of a stable laminar flow during injection into the microchannel [44].

The pioneering work has produced alginate micro-fibers via the microfluidic system. They have based their work on the coaxial flow, whereby the sheath flow (CaCl_2_) and the specimen flow (sodium alginate) meet at the intersection and leave the outlet in the form of continuous alginate micro-fibers [45]. This idea has been refined to make hollow alginate fibers by introducing a core fluid into the inner layer, resulting in the formation of a coaxial flow with three-layers: the sheath fluid, the specimen, and the core [46].

Compared to the wet-spinning and the electro-spinning methods, the microfluidic systems offer smoother and controllable micro-fiber manufacturing, making it possible to encapsulate cells within these fibers.

Figure 6 illustrates the concept of coaxial flow in the generation of continuous alginate micro-fibers [47,48].

### 4.2. Preparation of the Solutions

Alginate fibers are produced from alginate solution, extruded through a die immersed in a coagulation bath. Coagulation produces a physical hydrogel of alginate whose shape is determined by the extrusion die. The gel thus formed then undergoes various steps (drawing, drying, etc.) to form fibers.

Generally, alginate fibers are spun from an aqueous solution with a concentration between 4 and 10 wt% into a spinning bath. The content of the bath depends on the nature of these fibers to be spun. The coagulation bath usually contains calcium chloride, in the case of the preparation of fibers from calcium alginate. It is possible to use an acidic aqueous medium with calcium chloride and additives as a coagulation bath to improve the spinning process [49].

When spinning alginate acid fibers, the coagulation bath contains sodium sulfate and sulfuric acid. A 1–6 wt% sodium alginate solution can be extruded into a coagulation bath containing 0.2 M hydrochloric acid, depending on the desired viscosity [50]. After an appropriate immersion time, these alginate fibers are removed from the bath and washed in a water bath for approximately 20 s.

Calcium chloride (CaCl_2_) is the most commonly used reagent that allows for the ionic cross-linking of sodium alginate. The ionic gelation induced by calcium chloride is poorly controlled, irreversible, and rapid [51].

The coagulation bath may also contain a combination of salts: (copper, magnesium, and zinc salts) [52]. In the work of Qianqian Wang et al. [53], zinc (Zn^2+^), barium (Ba^2+^), copper (Cu^2+^), and aluminum (Al^3+^) ions were mixed with calcium (Ca^2+^) ions in the coagulation bath to improve the mechanical behaviors of these alginate fibers. A spinning solution was prepared by mixing 4 g of sodium alginate in 94 mL of distilled water. This solution was immersed in the coagulation bath, which contains the complex metal ions, calcium–zinc, calcium–aluminum, calcium–copper, and calcium–barium.

### 4.3. Effects of Process Parameters on Thermo-Mechanical and Physico-Chemical Properties

The structure of the fiber is highly dependent on the spinning conditions. In the case of wet-spinning, the low polymer concentration results in fibers that contain a large amount of solvent. The cross-sections and the structures of these fibers are influenced by the dehydration mechanism. In the work of B Niekraszewicz et al. [54], alginate fibers were fabricated using the wet-spinning technique. Morphological characterizations show that these elaborated fibers have porous structures and irregular cross-sections.

In their works, Su-Jung Shin et al. [55] elaborate continuous calcium alginate fibers with microfluidic devices. These authors investigated the variation of calcium alginate fiber diameters as a function of the sheath and sample flow rate. They showed that the fiber diameters increased almost linearly with increasing sample flow rates. The diameters of these elaborated fibers was regulated by changes in the sheath and sample flow rate.

Teresa Cuadros et al. [56] studied the impact of the residence time on the mechanical properties of alginate fibers. The concentrations of alginate and CaCl_2_ were fixed to 2 wt% and 0.5 wt%, respectively. They indicated that these elaborated fibers rapidly reach gelation (or equilibrium) with the surrounding solution.

The extrusion speed also has an impact on the properties of alginate fibers [57]. The cross-head speeds inferior to 10 mm.m^−1^ allows for an erratic alginate stream. In addition, there are several bumps and kinks along the gel of the alginate fibers. These resulting fibers always break at these bumps and kinks. For speeds greater than 20 mm.m^−1^, rapidly entangled coils are produced. Consequently, these authors used 15 mm.m^−1^ as the cross-head speed.

Magdalena Brzezińska et al. [58] studied the effect of the draw on the tenacity of alginate fibers. These authors showed that from a concentration of 12 wt%, alginate fibers resulted in a slightly higher tenacity (Table 1). It has also been shown that a drawing ratio of 100% gives a higher tenacity of around 15.54 cN·tex^−1^. These results can be explained by the greater orientation of the alginate macromolecules in the coagulation bath.

In another work, Lin H.Y. et al. [59] studied the impact of needle diameter and air pressure on the properties of the alginate fiber scaffold. They reported that from a pressure of 6 bar, the alginate solution was rapidly injected into the solution of the calcium. When the pressure has been decreased to 3 bar, the alginate solution flowed slowly into the calcium solution, allowing the formation of larger fibers. A slow injection speed allowed more time for the calcium ions to cross-link these alginate fibers.

In addition, it was shown that a larger alginate fiber was discovered when the diameter of the needle was augmented from 150 μm to 200 μm. A needle diameter of larger than 200 µm allowed the solution of alginate to seep out under the effect of gravity.

A higher water vapor transmission rate by alginate fibers was observed when increasing the air pressure and decreasing the needle size. These observations can be attributed to the enhanced adhesion between the contacting fibers, which generate a compact structure.

### 4.4. Effects of Solution Parameters on Thermo-Mechanical and Physico-Chemical Properties

Teresa Cuadros et al. [60] studied the effect of CaCl_2_ and sodium alginate concentrations on calcium alginate fiber properties. It was shown that when the concentration of CaCl_2_ was varied, the fiber diameters showed an oscillatory behavior. The amplitude of the diameter fluctuation was approximately 150 µm.

They showed that increasing the concentration of CaCl_2_ increased the tensile stress. In fact, a CaCl_2_ concentration of approximately 1.4 wt% allows for maximum fiber tensile stress. This effect can be justified by the fact that when the calcium ion concentration increased to a saturation point, the egg-box type sites were filled. A higher concentration of CaCl_2_ allows for a reduction in tensile stress, which is attributed to a partial collapse of the network. These results are in contradiction with the results of J. Zhang et al. [61], who showed that for higher concentrations of CaCl_2_, the tensile stress increased.

The pH of the solution also plays a role in the solubilization of alginates. A hydrogel can be formed when the pH of the alginate-containing solution is lower than the pKa of mannuronic acid (pKa = 3.38) and guluronic acid (pKa = 3.65). Each alginate has its own apparent pKa that depends on the distribution of G and M blocks on the chain, the alginate concentration, and the ionic strength of the aqueous solution [62].

A change in ionic strength in the solution can have a significant effect on the conformation of the polymer chain, and thus, on the viscosity of the solution. Furthermore, when the ionic strength increases, the solubilization rate of the alginate decreases. Thus, it is preferable to first solubilize the alginate in pure water before adding an ionic species under agitation [63].

It has been shown that the formation of hydrogels is due to the cross-linking of the carboxylate groups of the G residues with divalent cations. For this reason, a stiffer hydrogel can be formed by alginate fibers with high G, while a softer elastic hydrogel can be formed by fibers with high M [64].

In the presence of fluids, the alginate fibers with a high G content swell only slightly. Additionally, the structures of these fibers are not radically altered during processing. Calcium ions, Ca^2+^, are easily exchanged for sodium ions, Na^+^, when alginate fibers have a high M content. This allows these fibers to swell and become an elastic gel.

Yimin Qin et al. [65,66] have studied the properties of swelling of alginate fibers. The introduction of sodium ions into alginate fibers with high G content can regulate the swelling properties. The water absorption of calcium alginate fibers is lower than that of calcium–sodium alginate fibers. This is attributed to the ability of the sodium ions in these fibers to bind water.

Alginate fibers with a high G content and sodium–calcium alginate fibers are characterized by a lower salt solution absorption compared to fibers formed by calcium alginate with a high M content. These results can be attributed to the excellent gelling ability of fibers with high M loading [65].

Struszczyk H. [67] used a pilot-scale spinning apparatus to elaborate the alginate fibers. These fibers were obtained by immersing a sodium alginate solution in an acid coagulation bath. The impact of calcium ion concentration on the properties of the elaborated fibers was investigated (Table 2). Fibers with a moisture content of 16–26%, an elongation of 13–21%, and a tenacity of 16–23 cN·tex^−1^ were obtained when the substitution of the calcium ion was superior to 30%.

Rhoda Au Yeung et al. [68] elaborated calcium alginate fibers using two types of sodium alginate: alginate with a high G content and alginate with a high M content. Their results showed that the calcium content of alginate fibers with a high G content was 2.79 µmoles·mg^−1^, whereas alginate fibers with a high M content had only 2.58 µmoles·mg^−1^. This can be explained by the lower binding capacity of M residues compared to G residues.

According to mechanical tests, these authors indicated that there was no significant difference of the Young’s modulus between alginate with a high G content and alginate with a high M content. Indeed, the Young’s modulus value for alginate fibers with a high M content was approximately 5.63 GPa, and it was only 4.99 GPa for alginate fibers with a high G content.

It appears that the yield strength of alginate fibers with a high M content is approximately 15% higher, compared to alginate fibers with a high G content. This indicates that these fibers are resistant to irreversible deformation.

## 5. Alginate and Bio-Composites

Substances were mixed with alginate to improve the properties of alginate fibers. Alginate bio-composites are made by adding inorganic compounds such as hydroxyapatite (HA) and tetraethylorthosilicate (TEOS), synthetic polymers such as polypyrrole and polylactide, and natural polymers such as gelatin, chitosan, and collagen [69,70]. The mixing of other types of materials, such as bio-glass, ceramics, and inorganic materials based on carbon, and inorganic nano-particles, has also been studied [71,72].

### 5.1. Alginate–Polymer Blends

#### 5.1.1. Synthetic or Artificial Polymers

Weidong Zhou et al. [73] used polyethylene glycol diacrylate (PEGDA) as a filler in order to improve the fluidity of sodium alginate (SA). Their rheological results showed that the loss modulus G” and storage modulus G’ decreased with increasing PEGDA concentration. The decrease in G’ can be explained by the decrease in the degree of interaction between the SA molecular chains caused by the increase in molecular spacing. The decrease in G” can be attributed to the reduction in internal friction between the intermolecular force and SA molecules caused by PEGDA.

In another work, Wei-Wen Hu et al. [74] used electric field treatment to enhance gene transfection into alginate fibers reinforced by poly (caprolactone). It was found that the treatment enhanced the fluorescence intensity and the number of transfected cells, compared with the untreated group. These improvements were greater when the voltage values were smaller, to 1.5 V.

In order to facilitate the spinnability of an alginate bio-polymer, Xu, W. et al. [75] were used polylactic acid. These authors dissolved polylactic acid in chloroform and alginate in distilled water. Then, these two solutions were blended together to obtain emulsions. It was found that the tensile strength of the resulting fibers increased from 0.25 to 3.13 MPa.

Jie Liu et al. [76] developed sodium alginate fibers reinforced by cellulose nano-crystals, in order to enhance the mechanical behaviors of sodium alginate fibers. From the mechanical strength tests, these authors concluded that the mechanical properties of sodium alginate fibers were improved by the incorporation of cellulose nano-crystals. Indeed, the incorporation of cellulose nano-crystals increased the elongation at break and the tensile strength from 8.29% to 15.05% and from 1.54 to 2.05 cN·dtex^−1^, respectively. These observations can be attributed to the uniform distribution of the cellulose nano-crystals in the polymer matrix.

To ameliorate the flame-retardancy of alginate fibers, Xian-Sheng Zhang et al. [77] incorporated flame retardant viscose (FRV) into alginate fibers. The evolution of the heat release rate (HRR) and the total heat release (THR) are reported in Figure 7. These authors reported that alginate/FRV possessed a higher time to ignition (TTI). In contrast, bio-composite fibers showed lower HRR and THR values compared to neat alginate fibers. This effect may be related to the metal ions in alginate fibers that are considered to be flame retardant.

#### 5.1.2. Natural Polymers

Wawro, D. et al. [78] studied the structure of alginate fibers. Their results showed that alginate fibers have a porous structure and irregular cross-sections. To resolve this problem, these authors were used a chitosan bio-polymer in order to ameliorate the morphologies of alginate fibers. Indeed, significant differences in the cross-sections and fiber surface views are clearly visible when comparing alginate fibers and alginate/chitosan fibers. The flat indentations in the case of alginate/chitosan fibers are significantly smaller than in alginate fibers, and their cross-sections are more rounded.

Recently, Wang Bin et al. [79] have been successful in the elaboration of alginate/cotton blended fibers. Through morphological analyses, these authors have confirmed that the surfaces of alginate and cotton fibers are not very smooth. Additionally, the surface morphologies of alginate and cotton fibers are not regular cylindrical surfaces. In contrast, scanning electron microscope photographs of cotton/alginate blended fibers showed that the surface micro-morphologies between alginate fibers and cotton fibers are similar. In order to study the effect of chitin on the thermal properties of calcium alginate fibers, J. L. Shamshina et al. [80] examined the thermal properties of alginate/chitin fibers. Their thermo-gravimetric analysis results showed that the thermal stability of calcium alginate fibers was enhanced with the addition of chitin.

A few years later, Ma Xiaomei et al. [81] were used cellulose nano-crystals and its oxidized derivative as nano-fillers in order to enhance the flame retardant properties of alginate fibers. These author indicated that an enhancement in flame-retardancy was greater when using cellulose nano-crystals than its oxidized derivative. Indeed, the addition of a small amount of cellulose based nano-crystals decreased the limiting oxygen index of alginate composite fibers.

Furthermore, Wang Bin et al. [79] also studied the impact of alginate fibers on the flame retardant and combustion properties of alginate/cotton blended fibers (Figure 8). They concluded that alginate fibers improved the fire behavior and flame retardant properties of the elaborated alginate/cotton blended fibers.

In recent years, injectable hydrogels have attracted interest because of their capability to blend homogeneously with therapeutic agents and cells. Despite this advantage, the utilization of injectable hydrogels is nowadays severely restricted by the difficulty in improving bone regeneration, mimicking the natural environment of modified cells, and facilitating cell proliferation.

To remedy these problems, Bin Liu et al. [82] developed an injectable nanocomposite hydrogel composed of alginate reinforced by gelatin. These authors showed that the encapsulated rat bone marrow mesenchymal stem cells survived in the elaborated nanocomposite hydrogel. This study proves that the developed material can be used as a candidate for orthopedic applications.

### 5.2. Alginate/Nano-Particle Composites

#### 5.2.1. Zinc

Alginate fibers reinforced by nano-particles have recently received much attention. In studies by Andrea Dodero et al. [83], alginate fibers were loaded with zinc oxide nano-particles (ZnO-NPs). These nano-particles were produced using the sol–gel technique. The incorporation of ZnO-NP improved the rheological properties of the alginate. This is attributed to the electrostatic interactions and intermolecular hydrogen bonding between ZnO-NPs and the polysaccharide.

In addition, these authors indicated that it is preferable to use alginates with a high G content and a medium molecular weight, or with a high M content and a low molecular weight, when reinforced with ZnO-NPs. Indeed, a high G content allows for cavities along the chains of the polymer, which prevent the formation of interactions between these chains and the nano-particle molecules. Consequently, the impact of ZnO-NPs is nearly negligible. On the contrary, for M-rich alginates, the availability of establishing strong interactions with ZnO-NPs is important, due to the exposure of a high number of carboxyl and hydroxyl groups.

In another work, Guangyu Zhang et al. [84] coated calcium alginate nonwoven fabric with ZnO nano-particles, using the method of ion exchange.

Calcium alginate nonwoven fabrics were first immersed in Zn(NO_3_)_2_ solution, in order to obtain zinc calcium alginate fabrics. Indeed, the high-Zn^2+^ concentration solution allows part of the Ca^2+^ on the calcium alginate fibers to undergo an ion exchange reaction with Zn^2+^ (Figure 9a). Then, the zinc calcium alginate fabric was immersed in amino hyperbranched HBP solutions. Zn^2+^ was obtained in the solution after an ion exchange of Zn^2+^ with NH^3+^. A high temperature of 80 °C can convert Zn^2+^ into Zn(OH)_4_^2−^, and then, the ZnO-NPs are obtained. Finally, ZnO-NPs were bonded to the surface of calcium alginate fabrics (Figure 9b). Indeed, the force of attraction between the positive groups of the ZnO-NPs and the negative groups of the alginate fabric, and the interactions of the hydrogen bonds between the amino groups on the ZnO-NPS, and the hydroxyl and carboxyl groups on the fabric allow for the attachment of the ZnO-NPs on the alginate fabric.

#### 5.2.2. Silver

X. H. Zhao at al. [85] embedded alginate fibers with silver nano-particles (Ag-NPs) using the method of in situ reduction. First, alginate fibers and a silver nitrate (AgNO_3_) aqueous solution are mixed together. Ion exchange between the silver ions (Ag^+^) and the sodium or calcium ions in the alginate allows for the diffusion of Ag^+^ ions into alginate fibers. The fixation of Ag^+^ ions is performed thanks to an electrostatic attraction between the negative groups of the alginate and the positively charged Ag^+^ ions. The silver ions were then reduced in situ, so that the metallic silver generated can adhere to these elaborated fibers.

These authors indicate that in an aqueous medium, these elaborated fibers allow for the reduction of 4-nitrophenol 4-NP to 4-aminophenol 4-AP. The catalytic reduction is performed by the Ag-NPs by relaying the electrons from the BH_4_^−^ donor to the 4-NP acceptor (Figure 10).

In the first stage, metal hydride formation was achieved via the adsorption of BH_4_^−^ and its reaction with the surface of the elaborated alginate/Ag-NPs fibers (1) [86,87]. Due to the strong adsorption of the alginate/Ag-NPs fibers, 4-nitrophenol 4-NP can transport to the surface of the Ag-NPs (2). The desorption/adsorption equilibrium of the reactants on the surface of alginate/Ag-NPs fibers is fast. Then, the interaction of the adsorbed 4-NP with the silver nano-particles reduces the 4-NP [87]. The reduction reaction allows the formation of the 4-aminophenol 4-AP (3) [88]. A new reduction cycle (4) will take place when the 4-AP reactant is desorbed from the surfaces of the Ag-NPs [89].

In the work of Maila Castellano et al. [90], the alginate polymer was reinforced with Ag-NPs to elaborate nano-textured mats. According to morphological and spectroscopic tests, these authors confirmed that Ag-NPs were formed within an alginate polymer. The resulting material was then mixed with polyethylene oxide to produce alginate fibers using the electro-spinning technique. It has been demonstrated that these elaborated nano-fibers are insensitive to physical treatments. This allows for the use of ultraviolet light or heat as a sterilization method. On the other hand, basic or oxidizing reagents affect the stability of the prepared material, which confirms its sensitivity to chemical products.

Kaczmarek-Pawelska et al. [91] also studied the mechanical properties of alginate-based hydrogels reinforced by Ag nano-particles. These authors confirmed that the elaborated hydrogels are biomechanically compatible. Indeed, the obtained material showed mechanical properties very close to those of human skin. However, the increase in alginate concentration decreases the Young modulus. In fact, it is 8 MPa when the concentration of alginate is 0.1 mg/mL, whereas it is only 1.2 MPa when the concentration is 0.2 mg/mL.

#### 5.2.3. Graphene

Linhai Pan et al. [92] reinforced alginate fibers with graphene oxide as a filler in order to remove Cu^2+^ ions and Pb^2+^ ions from waste-water. Alginate fibers reinforced with graphene oxide nano-particles present a very high affinity with Pb^2+^ ions. The high adsorption for Cu^2+^ and Pb^2+^ is 102.4 and 386.5 mg·g^−1^. This high adsorption can be explained by the interaction of the oxygen atoms of alginate/graphene oxide fibers with the Cu^2+^ and Pb^2+^ ions. Based on the analysis of the adsorption mechanism, these authors confirmed that the chemical coordination and ion exchange effects (Figure 11) are responsible for the combination of heavy metals by alginate/graphene oxide fibers.

In other works, Xingzhu Fu et al. [93] used polymeric ionic liquids (PILs) to coat the surfaces of fibers based on calcium alginate using graphene, in order to develop conductive fibers.

To elaborate these fibers, these authors were used two coagulation baths. The first coagulation bath comprised calcium chloride. The second coagulation bath comprised a graphene aqueous dispersion. In fact, graphene aqueous dispersion was obtained by the dispersion of graphene by PIL.

In the step of secondary coagulation, the positively charged groups (imidazole ring) of LIP and the negatively charged groups (carboxylate ion) of alginate were linked together. The graphene layers attach to the surface of alginate fibers through the interactions of cation-π and π-π between graphene and PIL.

#### 5.2.4. Magnesium Oxide

In the work of De Silva et al. [94], alginate is reinforced with magnesium oxide nano-particles (MgO-NPs) with the aim of realizing nano-fibrous scaffolds. These authors have elaborated alginate fibers loaded by magnesium oxide whose diameters vary between 60 and 250 nm, using NPs of quasi-spherical shape. The tensile tests revealed that the mechanical behaviors of these obtained fibers were enhanced using the incorporation of MgO-NPs. Indeed, the reinforcement of alginate fibers by a 10 wt% of MgO-NPs resulted in a higher elastic modulus (*E*) and tensile strength (*σ_U_*) among the studied samples.

In order to elaborate the sodium alginate scaffolds loaded by MgO-NPs, Bijan Nasri-Nasrabad et al. [95] used a two-step technique: poly (vinyl alcohol) leaching and film casting. Their results showed that after the leaching step, the incorporation of 4 wt% MgO-NPs resulted in better mechanical properties (Table 3). Indeed, the incorporation of 4 wt% MgO-NPs improved the Young’s modulus of sodium alginate scaffolds by approximately 44%, compared to that of the neat sample. These improvements can be explained by the strong interaction between the alginate chains and the molecules of the nano-particles, and the decrease in the mobility of the polymer macromolecular chains.

Moreover, with an increase in the nano-particle content, the antibacterial properties of the scaffolds have been enhanced, with an increase in the concentration of the MgO-NPs (Table 3). With an introduction of 1 and 2 wt% MgO-NPs, the average diameter of the bacterial zone of the scaffold samples is more than 10 mm^2^, suggesting sensitive anti-microbial behavior. Whereas, with the incorporation of 3 wt% and 4 wt% MgO-NPs, the average diameter of the bacterial zone is reduced to less than 10 mm^2^, compared with the pure sodium alginate, which exhibits anti-microbial insensitivity. The improved antibacterial behavior can be explained by the anti-microbial effects of MgO-NPs protecting the obtained materials, in contrast to *P. aeruginosa* and *S. aureus*.

#### 5.2.5. Carbon Nanotubes

In order to have fibers with high electrical properties, Vijoya Sa et al. [96] prepared fibers based on alginate reinforced with carbon nano-tubes (CNTs) as a nano-filler, using a wet-spinning technique. This laboratory scale process could be used to produce industrial fibers via the addition of drawing/stretching steps. The use of calcium as a reticulation agent allows for electrostatic assembly between alginates and sodium dodecyl sulfate (SDS)-coated nano-tubes.

Both the alginate and the carbon nano-tubes are negatively charged. Thus, a repulsion will take place between the alginate solution and the aqueous solution of nano-tubes coated with SDS when they are added together. This prevents the agglomeration of the nano-tubes, and consequently allows for homogeneity in the spinning solution.

The interaction mechanism between CNT and the alginate is shown in Figure 12. The prepared solution is extruded into a coagulation bath containing an aqueous calcium chloride solution. Gel formation occurs when the spinning solution and the solution of calcium chloride are in contact. The coordination of Ca^+2^ ions in the cavities formed by the guluronate sequence pairs allows the formation of a calcium cage [97,98]. When the nano-tubes coated by SDS exist in the solution, Ca^+2^ ions link the nano-tubes and the alginate chains.

Recently, Aline Lima et al. [99] synthesized porous scaffolds based on CNT and hydroxamic alginate (HX). The HX was synthesized using the nucleophilic attack of hydroxylamine at the alginate carboxylic groups with dicyclohexylcarbodiimide. The partial modification of alginate with a derivative that is in acidic form facilitates its interaction with positively charged compounds.

From the study of mechanical behavior, these authors indicated that the HX/CNT scaffolds exhibited an improvement in their mechanical properties. According to FTIR and Raman spectroscopy, these authors confirmed an interaction between the alginate and the CNT cross-linked with the calcium.

#### 5.2.6. Hydroxyapatite

Fuqiang Wan et al. [100] elaborated alginate fibers loaded by hydroxyapatite (HAP) as a nano-filler, using the technique of spin-coating. These fibers are characterized by an anisotropic structure. The alignment of nano-filler wires and the formation of the anisotropic structure are achieved via mechanical force.

Gel formation occurs upon contact between the alginate spinning solution and calcium ions, Ca^2+^. Fiber formation takes place, after the covering of the glass substrate by the excess hydrogel (Figure 13).

Excess water is removed during the spin-coating step, followed by an additional gelling step. The mechanical stresses exerted by the centrifugal force and the incorporation of HAP nanowires are responsible for the formation of a fiber with interesting mechanical and physical properties.

In the research studies of Peilong Ni et al. [101] a composite fiber membrane based on sodium alginate/polyvinyl alcohol/hydroxyapatite was elaborated. In order to avoid precipitation and agglomeration of the HAP in the spinning solution, the latter was ultrasonically suspended in a solution, using an alginate polymer as a stabilizer. Their results showed that the distribution of HAP nano-particles is uniform with the absence of agglomerates when the concentration of alginate–hydroxyapatite in these fibers is augmented from 1.64 wt% to 6.25 wt%. In contrast, poor particle distribution with the appearance of agglomerates is observed when the alginate−hydroxyapatite loading is between 7.70 wt% and 9.10 wt%.

#### 5.2.7. Silica

Using the technique of microfluidic spinning, Zhang et al. [102] developed a novel fiber based on alginate reinforced by silica (SiO_2_). The morphology characterizations of these elaborated fibers were studied. They showed that alginate/silica composite fibers had a certain additive on the surface, whereas the alginate fibers had a smooth surface.

These authors also studied the performance of SiO_2_ nano-particles in improving the mechanical properties of alginate fibers. Indeed, they indicated that the addition of silica nano-particles improved the breaking stress of alginate fibers.

The application of alginate fibers is very limited in the biomedical field, because of their limited mechanical performance. To solve this problem, Lin Weng et al. [103] reinforced alginate fibers with SiO_2_ nano-particles using a microfluidic spinning technique. Mechanical tests showed that neat alginate fibers exhibited behaviors similar to brittle materials, with low elongation and stress. The addition of SiO_2_ nano-particles allows for the production of hybrid fibers with excellent mechanical performances, compared with the original alginate fibers. The elongation at break of the alginate fibers reinforced by SiO_2_ is 52.08%, whereas it is only 7.32% for the original alginate fibers. The breaking strength of the alginate fibers is augmented from 0.76 MPa to 4.96 MPa for the alginate/SiO_2_ fibers. These enhancements can be explained by the fact that the surface defects of alginate fibers are reduced by the SiO_2_ nano-particles.

## 6. Applications of Alginate

Nowadays, alginate fibers are widely used in various applications. Among the most important areas are cosmetic, hygiene, and medical textile materials, etc.

### 6.1. Cosmeto Textiles

Cosmeto textiles are materials with cosmetic behaviors. However, these textile materials can also have other functions, such as UV protection agents, medical properties, and odor reducers.

Cosmetic textiles are an industry developed to ensure the well-being and health of consumers. Textile fibers are used to deliver a large variety of microencapsulated ingredients such as vitamin E, aloe vera, caffeine, retinol, etc. [104]. The new generation of cosmetic textile products uses innovative new techniques to provide medical, anti-aging, and stress relief benefits through clothing, textiles, and other products. In this regard, alginate fibers are highly biocompatible and hydrophilic, making them ideal products for the elaboration of face masks. In addition, alginate fibers can be used to carry several bioactive substances, allowing for sustained release on the skin.

### 6.2. Waste-Water Treatment

The elimination of dye molecules from waste-water is a complicated mechanism because of their inertness [105]. The active hydroxyl and carboxyl groups of alginate fibers have been investigated for the removal of dye molecules from effluents [75]. Electro-spun alginate fibers show interesting properties such as a high specific surface area and porosity. These properties make these materials suitable for waste-water treatment. A recent study by Zhao, X. et al. [106] demonstrated that fibrous sodium alginate/chitosan composite foam presents the potential to eliminate anionic and cationic dyes from waste-waters (Figure 14). Its high adsorption capacity can be attributed to its inter-connective pores and microscale fibers. According to the adsorption kinetics, these authors indicated that the adsorption rates of anionic and cationic dyes were initially rapid and then progressively slowed down to equilibrium.

Additionally, alginate fibers were used to prepare nano-fibrous membranes for metal adsorption. Mokhena et al. [107] fabricated alginate fibers using polyethylene oxide as the carrier polymer for copper adsorption. They found that these obtained fibers were characterized by a high porosity and a large surface area. These properties allow these macro-porous fibrous membranes to exhibit an excellent capacity for removing copper ions from the aqueous medium. The prepared membrane can also recover nickel and copper ions, since it shows better selectivity upon these ions. In other research works, the authors coated alginate fibers with cellulose, in order to eliminate chromium from effluents [108]. It was shown that these prepared fibers for a greater than 80% rejection of chromium ions. These observations can be attributed to the existence of the carboxyl and hydroxyl groups of alginate, as well as the hydroxyl and sulfate groups of cellulose.

In the works of F. Sun et al. [109], alginate fibers were used as filtration membranes in order to separate oil from water. These authors modified the surface properties of alginate fibers by incorporating acrylonitrile into alginate fibers. They obtained fibers with a high oil affinity. It was found that the angle of contact with the water was augmented from 56° to 70° with increasing acrylonitrile content. These effects can be explained by the strong interaction between molecular chains in the developed material, due to the existence of highly polar -CN substituents. The hydrophilic properties of calcium Ca^2+^ ion cross-linked alginate fibers have been examined for the retention of oil by Mokhena and co-workers [108]. 

### 6.3. Wound-Dressing

Wound-healing, which is one of the most complicated processes, involves a series of events, such as cell response, growth, and differentiation [110]. Consequently, the products used for the treatment of wounds must be characterized by durability, non-toxicity, and flexibility. Moisture-regulating and oxygen-transporting porosity is one of the supplementary properties conferred by the electro-spinning technique for wound-healing. Additionally, the production of multifunctional materials through the incorporation of bioactive compounds facilitates wound-healing [111].

Alginate is one of the most commonly used bio-polymers for dressing products, thanks to its interesting properties such as biodegradability, bio-compatibility, great absorption capacities, low toxicity, and low cost [112]. Alginate fibers have ion exchange behaviors when in contact with wound exudates. In fact, calcium ions are replaced by sodium ions from the body fluid, allowing for the development of a moist gel on the wound surface. The addition of other compounds to alginate fibers offers the possibility for producing advanced materials with several advantages such as a gel-forming ability and hemostatic capability. Alginate fibers loaded with chitosan for cancer stem-like cell enrichment were developed by Kievit’s group [113,114]. They mixed chitosan and alginate and freeze-dried them in order to elaborate a porous sponge. The obtained materials allowed the enrichment of cancer stem cells by hepatocellular carcinoma and glioma-stoma. Nevertheless, the ratio of the composition is very limited, due to the restricted compatibility between the compounds. Tumor niches produce external factors that control the fate determination and the numbers of the stem cells. For this reason, it is necessary to prepare new materials allowing for the customization of the scaffold behaviors for cancer stem cells.

In this context, Wei-Wen Hu et al. [115] elaborated alginate fibers loaded with poly(caprolactone) using the method of co-electrospinning. They concluded that a low ratio of poly(caprolactone) considerably enhanced the cancer stem cell behaviors of these elaborated fibers. This can be justified by the spatial separation of cell populations by the sparse poly(caprolactone), which allows for the concentration of the cancer stem cells. In addition, it was found that neat alginate fibers and alginate fibers reinforced by poly(caprolactone) significantly decreased the wound area compared to tissue culture polystyrene and poly(caprolactone).

According to wound-healing studies, J. L. Shamshina et al. [80] indicated that the addition of chitin to calcium alginate fibers accelerated wound closure. It has been found that the wound sites covered by calcium alginate/chitin fibers have undergone normal wound-healing.

Yimin Qin [116] incorporated silver (Ag) into alginate fibers in order to ameliorate their anti-microbial properties. It has been demonstrated that alginate fibers reinforced with Ag nano-particles can be used to make highly absorbent and anti-microbial dressings. This effect can be explained by the liberation of Ag ions used as nano-fillers, which have anti-microbial properties. Bacteria trapped in alginate dressings will be killed by the Ag ions (Figure 15).

Additionally, alginate-based hydrogels reinforced with Ag nano-particles have been used for wound-healing in various animal models, and they prevent contamination. Alginate nanocomposite hydrogels showed antibacterial activity in the long term, and sustained Ag release [117]. In the same context, Diniz, F.R. et al. elaborated alginate/gelatin hydrogels reinforced with Ag nano-particles for wound-healing. The obtained results showed that the elaborated product is characterized by antibacterial activity against *P. aeruginosa* and *S. aureus*, and is non-toxic against fibroblasts [118].

### 6.4. Tissue Engineering

Tissue engineering aims to replace, maintain, or improve the function of human tissues, thanks to tissue substitutes. It is therefore a matter of elaborating artificial tissues, using cell cultures, biomaterials, and growth factors, in order to obtain a hybrid biomaterial.

Alginate gels have many applications in tissue engineering. Indeed, alginate is a non-toxic, inert, and non-immunogenic substance which thus presents the required characteristics to constitute a good scaffold for tissue engineering [119].

The applications of alginate fibers in tissue engineering are very limited according to cell adhesion and viability. The addition of bioactive substances onto alginate fibers in order to ameliorate cell adhesion and proliferation makes these fibers ideal materials for the elaboration of scaffolds.

Jeong S. I. et al. [120] developed alginate fibers covalently bonded with a cellular adhesive in order to improve cell growth and viability. The results of these authors show that the addition of an adhesive peptide improved the propagation and adhesion of cells without changing the morphology of these fibers. In another study, Jeong S. I. et al. [121] elaborated a polyionic complex based on alginate and chitosan to achieve cell attachment and proliferation. The swelling rate in the deionized medium decreased with an increase in the concentration of chitosan content. Mouse pre-osteoblastic cells adhered to the alginate/chitosan nano-fibrous membrane and showed substantial proliferation.

To combine the properties of alginate and chitosan, Xinxin and Christopher [122] developed a process to treat the surface of alginate fibers with an aqueous solution of chitosan. After the absorption of chitosan on the surface of the calcium alginate fibers, these alginate/chitosan fibers are then freeze-dried. These fibrous materials are subsequently used as scaffolding materials in tissue engineering. Figure 16 gives us an idea regarding the applications of alginate/chitosan fibers in the area of tissue engineering.

In addition, the reinforcement of alginate fibers resulted in the improvement of mechanical behaviors, and therefore, in the usefulness of these materials for tissue engineering. In their studies, Tao, F. et al. [123] prepared fibers based on sodium alginate, carboxy-methyl chitosan, and biodegradable polymer poly(caprolactone) via the electro-spinning technique. According to tensile test analyses, the tensile strength of the elaborated micron-fibers was significantly higher than those of the poly(caprolactone)/carboxymethyl chitosan and poly(caprolactone)/sodium alginate micron-fibers. Further, these authors concluded that these obtained fibers can be used for periosteal tissue engineering.

In another work, alginate/chitosan–polylactide fibers were elaborated by Wu Hua et al. [124] for applications of neural tissue engineering. The mechanical properties of dry chitosan fibers showed that the modulus, tenacity, and elongation were around 25 cN·dtex^−1^, 1.5 cN·dtex^−1^, and 10%, respectively. The chitosan–polylactide fibers showed a higher modulus, tenacity, and elongation in the dry and wet states compared to the chitosan fibers. These results can be attributed to the polylactide component in chitosan–polylactide fibers, which is hydrophobic and mechanically strong. In addition, it found that the modulus and the tenacity of the alginate/chitosan–polylactide fibers are higher than that of chitosan–polylactide fibers in the dry state, whereas the modulus, tenacity, and elongation are still similar in the wet state.

Alginates are also reported as natural polymers utilized in hydrogel-based nanocomposites. A technique widely used in the elaboration of alginate-based nanocomposites is the chemical modification of the polymer in order to improve the interaction between the polymer matrix and the nano-particles, and thus to form a hydrogel characterized by stable mechanical properties. Nevertheless, chemical modification of the natural polymers can alter their biocompatibilities, which is a reason for using unmodified bio-polymers.

In this context, Rebeca Leu Alexa et al. have studied the different manufacturing methods for obtaining alginate–natural clay hydrogel-based nanocomposites adapted to 3D printing processes. The 3D multilayered scaffolds were obtained by printing the nanocomposite inks using the extrusion technique. The properties of the obtained materials confirm their use in tissue engineering. According to the biological analysis, these authors showed that the addition of unmodified clay into the alginate polymer allows for the development of cells [125].

### 6.5. Anti-Microbial Activity

In their work, Dumont et al. [126] studied the antibacterial activity of alginate-reinforced chitosan fibers, which were prepared using the technique of wet-spinning.

It was found that the inclusion of chitosan offers antibacterial properties, and that alginate gives healing properties and good hemostatic properties to these elaborated fibers (Figure 17). They concluded that the addition of chitosan on alginate fibers provides antibacterial activities contrary to Escherichia coli, Staphylococcus epidermidis, and various strains of *Staphylococcus aureus*, namely Healthcare Associated Methicillin Resistant *Staphylococcus aureus* (HA-MRSA), Methicillin Sensitive *Staphylococcus aureus* (MSSA), and Community Associated Methicillin Resistant *Staphylococcus aureus* (CA-MRSA).

Other authors such as Sibaja Bernal et al. [127] indicated that alginate/chitosan fibers showed an excellent degree of inhibition of *Escherichia coli* growth, according to the test of bacterial inhibition. In fact, a great bacterial growth inhibition area was observed around these fibers loaded with the sulfathiazole drug after 24 h of incubation at 37 °C.

In the studies of Batista, M. P. et al. [128], a new route towards hybrid alginate/chitosan fibers via the emulsion gelation technique was developed. According to the standard tests, these authors indicated a clear antibacterial activity of these alginate/chitosan fibers against *Klebsiella pneumonia* and *Staphylococcus aureus*. To achieve this study, these authors were used two different methodologies with various contact times between the selected bacterial inocula and the specimens.

Recently, there have been several studies that investigate the antibacterial properties of alginate-based composite hydrogels. In general, changes in the structure of chitosan decrease its anti-microbial properties [129]. The incorporation of alginate hydrogels into chitosan solution showed a greater than 99% anti-microbial activity, as compared to the neat alginate hydrogel. Additionally, the addition of the chitosan into alginate hydrogels enhanced their anti-microbial properties [130].

It has been shown that the introduction of a anti-microbial peptide into sodium composite hydrogels based on alginate/polyethylene glycol provides good biocompatibility and an improvement in antibacterial activity [131]. A hydrogel loaded with an ultrashort peptide has been formulated for the treatment of the eyes and skin infections, and for the prevention of biomaterial infections [132]. An amphiphilic anti-bacterial hydrogel has been developed for skin wound treatments, and shows antimicrobial properties against bacteria, such as *S. aureus* and *E. coli* [133].

### 6.6. Sensors and Energy

Fibers elaborated using the electro-spinning method have interesting characteristics such as a large surface area, ease of modification of surface functionality, and malleable mechanical properties. These unique characteristics allow these fibers to provide a novel platform for the design of new sensors with high portability, sensitivity, and selectivity [134,135]. In order to improve the sensitivity, the response time, and the detection level, many sensing agents have been incorporated into electro-spun fibers. Alginate has been functionalized with heavy metal-sensitive compounds. Its carboxyl and hydroxyl groups can bind to multivalent ions, allowing them to be easily detected [135,136]. For this purpose, alginate fibers can be labeled with fluorescent sensors that are greatly selective in detecting metal ions found in aqueous solutions [136].

Wei-Peng Hu et al. [137] reinforced alginate fibers with silver nano-particles to obtain moisture-sensitive materials for respiratory sensors to monitor breathing during exercise and changes in emotion. It was demonstrated that the obtained material was able to detect the respiratory rhythm during a race, by fixing it on the seal of the mask exhalation valve. Additionally, these authors showed that the masks are characterized by stability and reusability, because they yielded the same results after 3 months. In the case of changes in emotion, it was shown that the mask was able to distinguish between sadness and pleasure by monitoring breathing frequencies.

Chen, H. et al. [138] developed alginate fibers reinforced with gels for the preparation of skin intelligence, such as ionic sensors. It was found that the gels maintained good electrical conductivity and mechanical deformability when exposed to long-term storage under ambient conditions or extreme conditions. The gels could be stretched and knotted without any damage to the structure. After storage for 6 h at 40 °C or −18 °C, they were capable of illuminating an LED bulb (Figure 18a). Then, the change in the conductivity of the gels was discovered in the range of the temperature between −20 °C and 40 °C, or at 25 °C for 6 days (Figure 18b,c).

In research by Ying He et al. [134], the authors elaborated fluorescent fibers based on alginate reinforced using gold nanoclusters and chicken egg white using the wet-spinning technique. These prepared fibers present a fluorescent sensor of great selectivity for detecting Hg^2+^ and Cu^2+^ ions among several metal ions in an aqueous solution. Among 11 kinds of common cations, Hg^2+^ cations could completely extinguish the fluorescence, while Cu^2+^ cations caused an obvious decrease in the intensity of the fluorescence.

In the same focus, these authors used these fluorescent fibers as an anti-counterfeiting label into cotton textiles, using the knitting technique. These smart fibers can be used in the design of novel flexible optical sensors and wearable optical sensors.

Another pressure sensor characterized by interesting sensing and mechanical properties has been developed from a composite hydrogel based on sodium alginate/polyacrylamide nanofibrils [139]. In fact, the compression, tension strength, stretchability, toughness, and elasticity of the obtained material are 4 MPa, 0.750 MPa, 3120%, 4.77 MJ m^−3^, and 100%, respectively. The ionic conductors have been presented as sensors with deformation variations of between 0.3 and 1800%, a low applied voltage of up to 0.04 V, and a high sensitivity to pressure, equal to 1.45 kPa^−1^. These ionic sensors could be used in sports tracking applications, soft robotics, and machine/human interfaces.

### 6.7. Antiviral Activity

According to the antiviral performance of ionic polysaccharides, like alginate, they will play significant roles in the anti-COVID-19 field [140]. These bio-polymers are able to cause a slow release, prevent antigen degradation, and improve their stability, thereby enhancing immunogenicity. They can interact directly with the surface of viruses, and inhibit their infectivity or murder them [141].

Dental impressions, dentures, occlusal records, and trays can be contaminated with viruses and bacteria. The studies carried out do not prove the contamination or survival of the virus and the dental impression; however, salivary contamination suggests the possibility of the presence of a viral biological load in addition to those of yeasts and bacteria [142].

Generally, principal recommendations are linked to the danger of infections when a dental impression is carried out using conventional procedures. Nevertheless, there are no contamination or specific notes on disinfectants for tips, impression, scanner, cast, and hardware for computer aided manufacturing/computer aided design [143]. It can be noted that when taking dental impressions using both digital and traditional procedures, dental personnel are at risk of danger, due to close contact with the droplets and the aerosols of patients; however, they experience a different degree of exposure to aerosol-generating procedures and oral fluids [144].

The main advantages of digital techniques are the close-contact minimization of dental personnel with patients, and limited transmission through aerosol-generating procedures and respiratory droplets. These features are very essential for COVID-19 prevention, and particularly for dental care [145].

It is recommended that fewer objects are left on surfaces, in order to decrease the possibility of contamination of the surfaces, equipment, and environment. Computer keyboards must be covered with films based on polyethylene. Additionally, surfaces contaminated with biological particles should be disinfected using appropriate detergents [146]. According to in vitro studies, the chemical disinfection of alginate by sodium hypochlorite, glutaraldehyde, alcohol, and chlorhexidine reduced microbial counts on the surface without altering the dimensional stability of alginate impressions. Therefore, these disinfectant agents could be exploited to decrease the cross-contamination of alginate impressions [147,148].

## 7. Conclusions and Future Perspectives

In this work, we gave the state of knowledge regarding the definition, source, structure, and specific properties of alginate bio-polymers. Additionally, we have detailed the different strategies of alginate spinning and the influences of processes and solution parameters on the properties of alginate fibers. The article also discusses the influence of a wide range of materials on the properties of the obtained fibers from these bio-polymers. Finally, the potential applications of these bio-composite fibers of alginate are discussed.

Alginates are ecologically and environmentally friendly. The particular properties of these bio-polymers allow fibers produced from them to find more and more uses in special applications, especially for medical uses. Alginate fibers are commonly made by extruding a sodium alginate solution into a calcium chloride bath, producing calcium alginate fibers. Calcium alginate fibers should have a high potential for some specific applications. The incorporation of fillers into calcium alginate fibers seems to be one of the most successful solutions. These fillers allow for the improvement of the physical, thermal, mechanical, and wound-healing properties of calcium alginate fibers.

Although the spinning of alginate fibers and the reinforcement of these fibers by substances of different natures in order to enlarge their fields of application have been widely studied over the last few decades, there are still other important topics that deserve to be further investigated. For example, the different techniques of the mixed spinning of alginate and chitosan fibers. In addition, the properties of these mixed fibers and their applications in various potential applications also deserve further analysis. All of this remains within the framework of encouraging researchers and industries to develop innovative and sustainable materials.

## Figures and Tables

**Figure 1 jfb-13-00117-f001:**
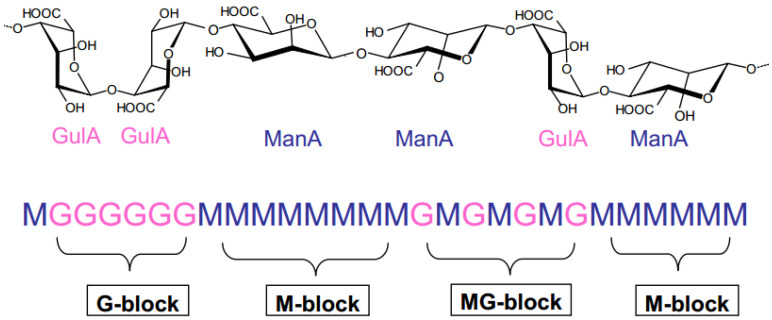
Alginate structure.

**Figure 2 jfb-13-00117-f002:**
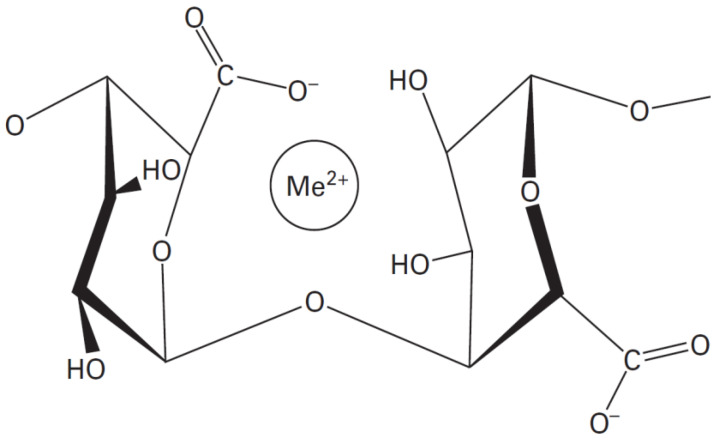
The stereo-chemical structure of the GG block [25].

**Figure 3 jfb-13-00117-f003:**
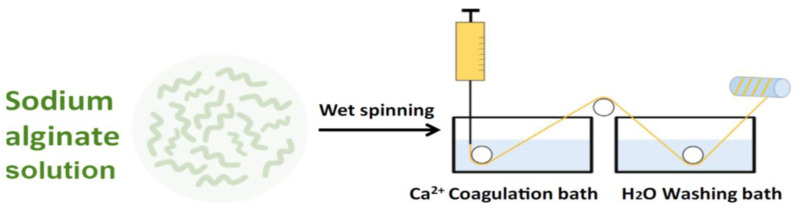
Schematization of the wet-spinning process for preparing alginate fibers.

**Figure 4 jfb-13-00117-f004:**
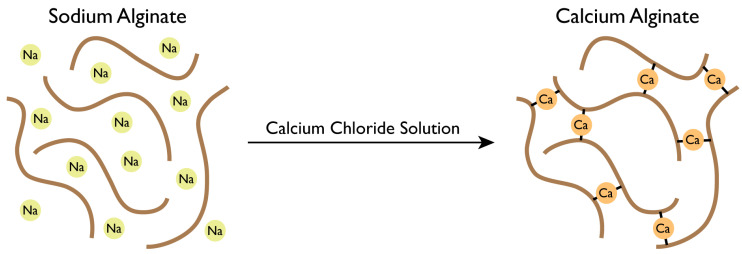
Gellification of sodium alginate solution.

**Figure 5 jfb-13-00117-f005:**
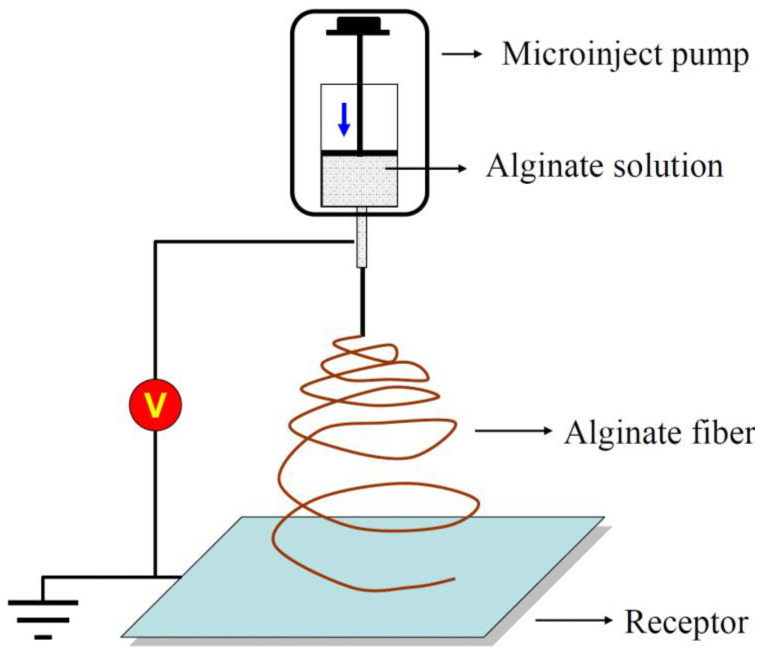
Illustration of the electro-spinning process for preparing alginate fibers.

**Figure 6 jfb-13-00117-f006:**
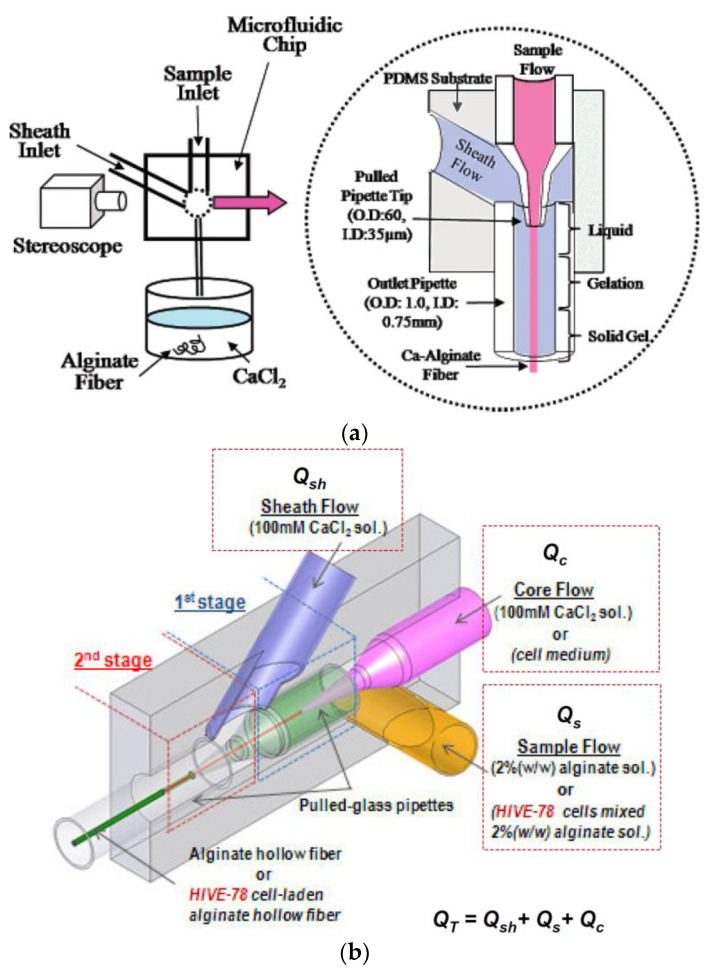
Microfluidic concept for (**a**) continuous alginate micro-fibers and (**b**) continuous alginate hollow micro-fibers.

**Figure 7 jfb-13-00117-f007:**
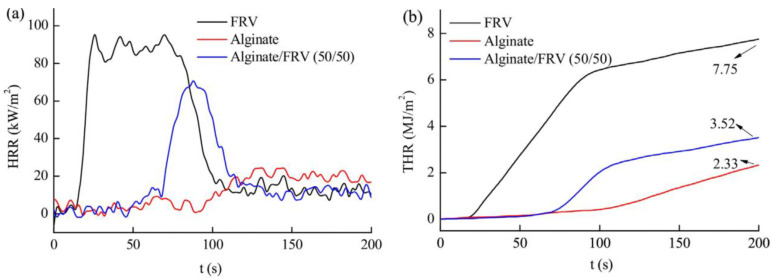
Evolution of (**a**) HRR and (**b**) THR as function of time of prepared fibers [77].

**Figure 8 jfb-13-00117-f008:**
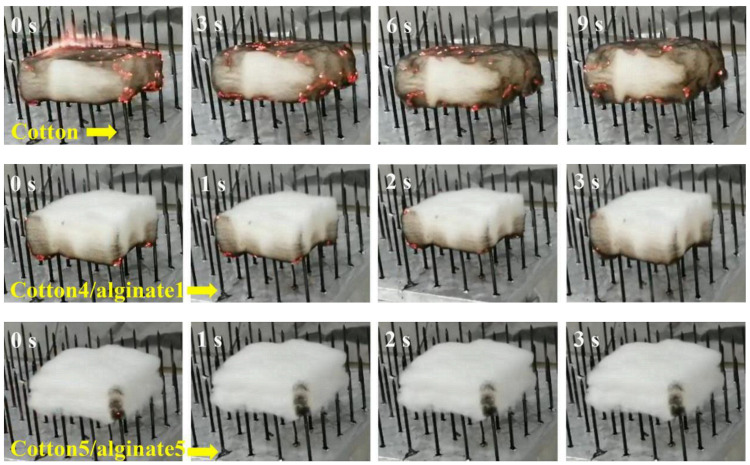
Photos of a flammability test for specimens recorded at several time points [79].

**Figure 9 jfb-13-00117-f009:**
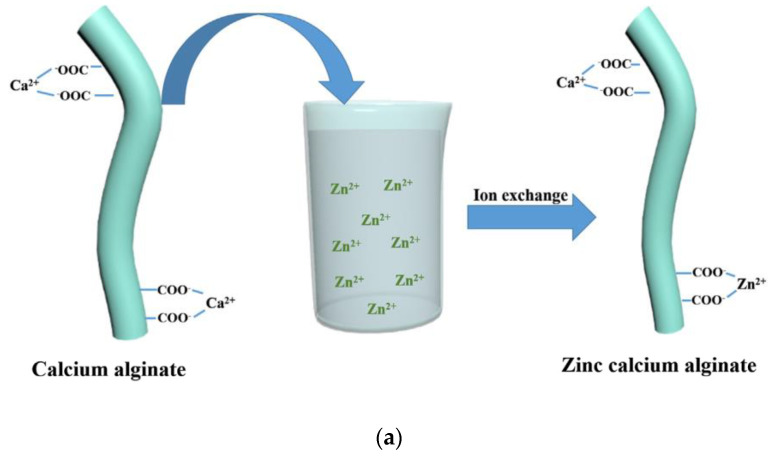
The coating of ZnO−NPs on an alginate fabric [84]. (**a**) Ion exchange reaction between Ca^2+^ and Zn^2+^; (**b**) Synthesis of the ZnO NP−coated alginate fabric.

**Figure 10 jfb-13-00117-f010:**
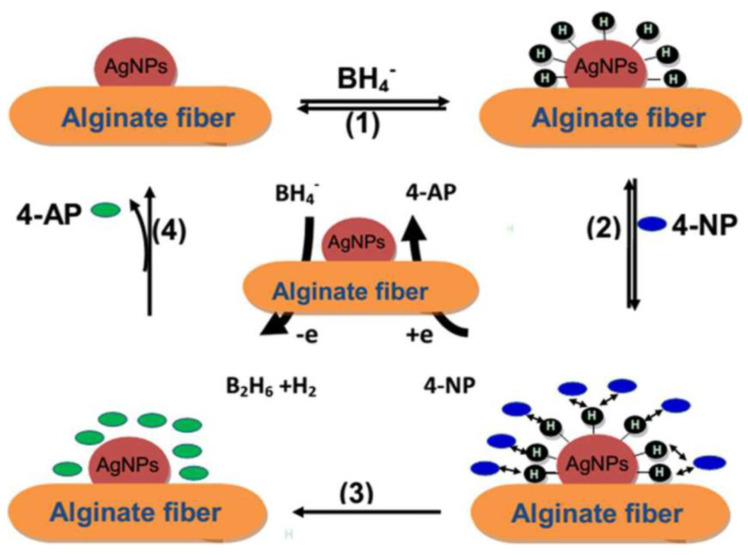
The mechanism of 4−NP reduction in the presence of the alginate/Ag−NP fibers [85].

**Figure 11 jfb-13-00117-f011:**
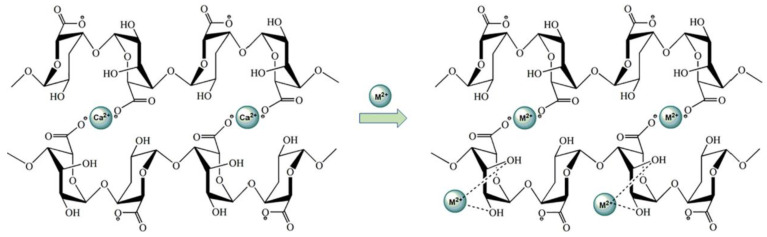
Mechanism interaction between alginate/graphene oxide fibers and heavy metals [92]. M^2+^: Cu^2+^ or Pb^2+^ ions.

**Figure 12 jfb-13-00117-f012:**
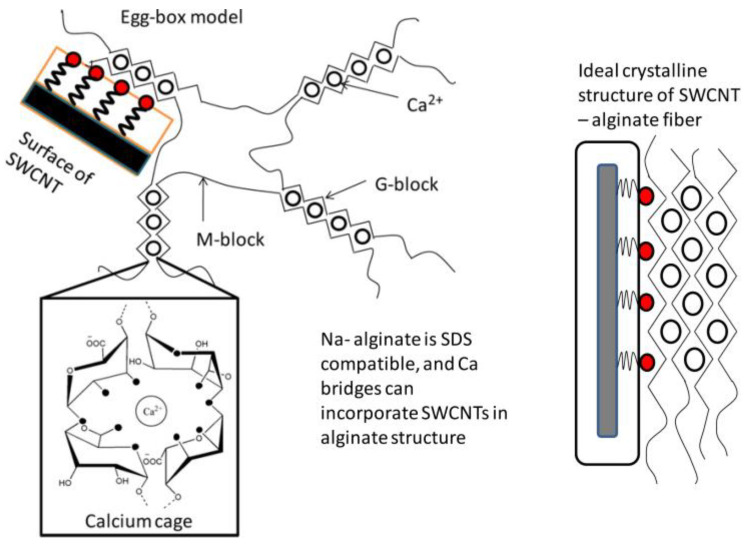
The interaction mechanism between CNT and alginate [96].

**Figure 13 jfb-13-00117-f013:**
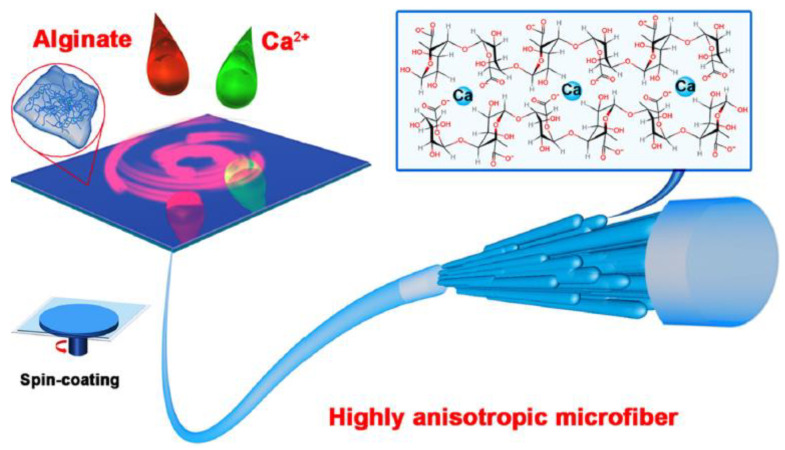
Elaboration of alginate fibers with anisotropic structure [100].

**Figure 14 jfb-13-00117-f014:**
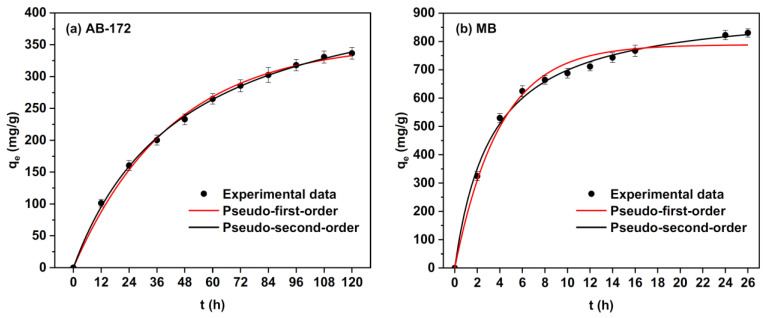
Adsorption kinetics, pseudo-first-order, and pseudo-second-order non-linear fitting curves of MB and AB-172 on the elaborated composite foams [106].

**Figure 15 jfb-13-00117-f015:**
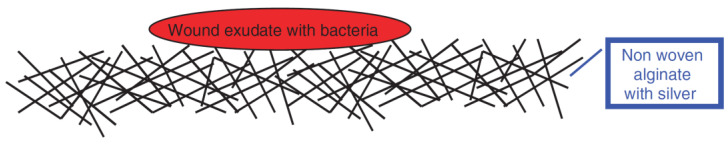
The anti-microbial mechanism of the elaborated wound-dressings [116].

**Figure 16 jfb-13-00117-f016:**
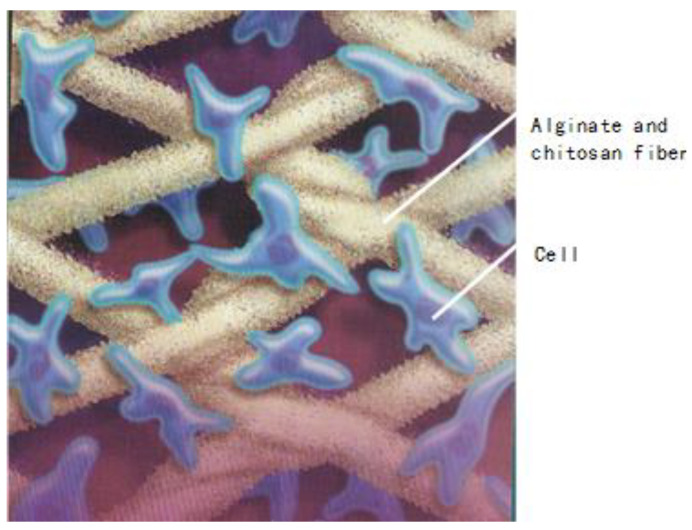
Chitosan and alginate fibers and their applications in tissue engineering.

**Figure 17 jfb-13-00117-f017:**
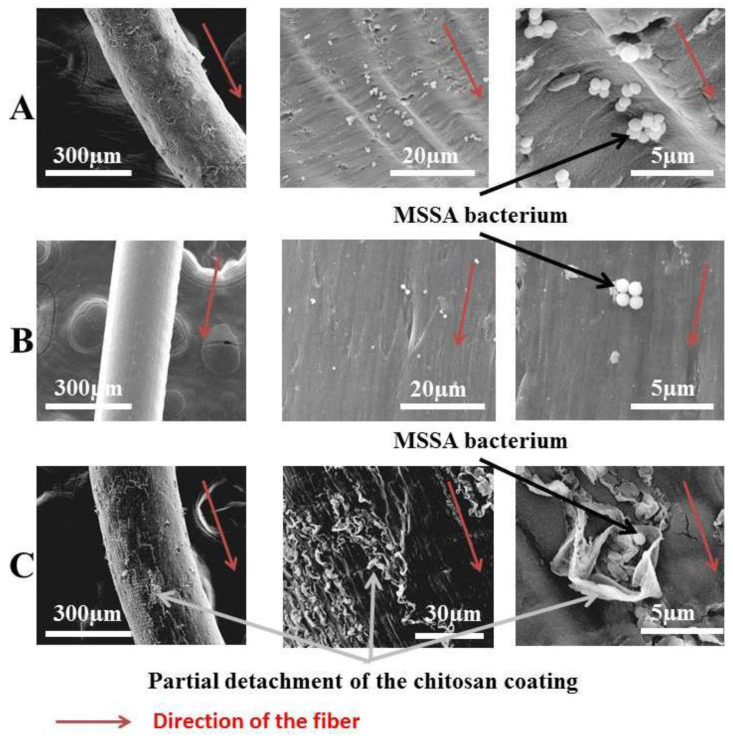
Scanning electron microscope observation of alginate fibers (**A**), chitosan fibers (**B**), and alginate-reinforced chitosan fibers (**C**) [126].

**Figure 18 jfb-13-00117-f018:**
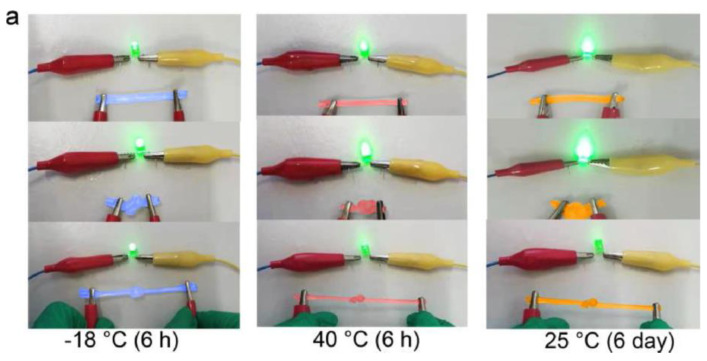
Stretched and knotted gels illuminated an LED bulb (**a**); The conductivity changed with temperature (**b**); The conductivity changed with storage time (**c**) [138].

**Table 1 jfb-13-00117-t001:** Conditions of fiber formation and alginate fiber properties.

Samples	Polymer Content (wt%)	Drawing Ration (%)	Total Drawing Ratio (%)	Tenacity (cN·tex^−1^)	Elongation at Break (%)
S1	12	50	70	14.66	5.37
S2	12	100	70	15.54	4.89
S3	13	50	74	14.39	4.2
S4	13	100	70	14.59	5.36

**Table 2 jfb-13-00117-t002:** Impact of calcium ion concentration on alginate fiber properties.

**Calcium Ion Concentration (wt%)**	**Substitution Degree (%)**	**Water Retention Value (%)**
3.2	31	561
10.2	99	97

**Table 3 jfb-13-00117-t003:** Mechanical and bacterial behaviors of the sodium alginate scaffolds loaded with different MgO-NP concentrations.

**Nano-Particles Concentration (wt%)**	**Young Modulus (kPa)**	**Average Diameter of the Inhibitory Zone (mm^2^)**
0	180.4 ± 15.2	13.6 ± 1.8
1	190.5 ± 25.2	13.1 ± 2.1
2	230.1 ± 27.8	11.7 ± 1.3
3	250.8 ± 30.4	9.8 ± 1.7
4	260.3 ± 19.6	9.6 ± 1.9

## Data Availability

Not applicable.

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
