# Peer review of "Alginate-Based Bio-Composites and Their Potential Applications"

_jfb, 2022, doi:10.3390/jfb13030117_

Round 1
Reviewer 1 Report
Alginate is a type of biomaterial from natural resources, which is widely applied as workhorse in various applications. I think the first and second sections are nice, but the third and fourth parst in this review just focus the ALg-based fibers, not almost of bio-composites as presented in the title. for example, ALg-based nanocomposite hydrogels or organogels are reported and used in literatures. The authors did not overview the advances of Alg-based biomaterials in a comprehensive manner, and References listed in this manuscript are also not high-impact Journals. I do not think this is a suitable way to write a good review, inversely, this is likely to a summary of their own work. We suggest the authors need to replenish more advanced Alg-based composite systems and recent high-level references, more importantly, have better to improve this manuscript in a more comprehensive and accurate way.
Author Response
Response to Reviewer 1 Comments
Manuscript Title : Alginate based bio-composites and their potential applications
Manuscript ID : jfb-1837510
Modifications according to reviewer 1 comments are presented in red color in revised manuscript.

Reviewer 2 Report
The present review paper is interesting and describe very well the state of knowledge and different strategies for alginate fibers process and their potential applications.
They have only a few corrections to be published.
Page 15, in 5.1.1 said polyethylene glycol acrylate, but PEGDA is polyethyleneglycol diacrylate.
Page 17, 5.2.1 say have recently garnered, what is it garnered?
Fig 18. It didn´t explain well the meaning of change of colors and length changes, it is necessary to clarify.
Author Response
Response to Reviewer 2 Comments
Manuscript Title : Alginate based bio-composites and their potential applications
Manuscript ID : jfb-1837510
Modifications according to reviewer 2 comments are presented in green color in revised manuscript.

Reviewer 3 Report
Dear Authors
the paper is interesting and can be considered for publication
however, some clinical topics are needed before acceptance
in particular:
- the chemical and physical properties of alginate can favour covid-19 diffusion when used in patients for dental impressions? How its use in the light of covid19 indications? Please cite PubMed ID33135082
- In the same way, what about potential aerosol/microparticles dispersion according to alginate properties when a dentist takes an impression? Can be useful - to diminish microparticles contamination - the use of air purifier systems. Please cite PubMed ID35564533
Author Response
Response to Reviewer 3 Comments
Manuscript Title : Alginate based bio-composites and their potential applications
Manuscript ID : jfb-1837510
Modifications according to reviewer 3 comments are presented in blue color in revised manuscript.

Round 2
Reviewer 1 Report
The authors have addressed all my concerns related to this manuscript, I think the reversion is acceptable for this journal